

# Brain derived neurotrophic factor declines after complete curative resection in gastrointestinal cancer

Tomasz Guzel[1], Katarzyna Mech[1], Marzena Iwanowska[2], Marek Wroński[1] and Maciej Słodkowski[1]

[1] Department of General, Gastroenterology and Oncologic Surgery, Medical University of Warsaw, Warsaw, Poland
[2] Department of Laboratory Diagnostics, Medical University of Warsaw, Warsaw, Poland

Corresponding author
Tomasz Guzel,
tomasz.guzel@wum.edu.pl

## ABSTRACT

**Background:** Brain derived neurotrophic factor (BDNF) is a neurotrophin involved in neural and metabolic diseases, but it is also one of the crucial factors in cancer development and metastases. In the current study, we investigated serum BDNF concentrations in patients that underwent surgical treatment for colorectal cancer or pancreatic cancer.

**Methods:** Serum BDNF concentrations were measured with standard enzyme-linked immunosorbent assays, before and on the third day after the operation, in 50 consecutive patients with colorectal cancer and 25 patients with pancreatic cancer (tumours in the head of pancreas). We compared pre- and postoperative BDNF levels, according to the subsequent TNM stage, histologic stage, lymph node involvement, neuro- or angio-invasion, and resection range.

**Results:** In the pancreatic cancer group, BDNF concentrations fell significantly postoperatively ($p = 0.011$). In patients that underwent resections, BDNF concentrations fell ($p = 0.0098$), but not in patients that did not undergo resections (*i.e.*, laparotomy alone). There were significant pre- and postoperative differences in BDNF levels among patients with ($p = 0.021$) and without ($p = 0.034$) distant metastases. Significant reductions in BDNF were observed postoperatively in patients with small tumours (*i.e.*, below the median size; $p = 0.023$), in patients with negative angio- or lymphatic invasion ($p = 0.028$, $p = 0.011$, respectively), and in patients with lymph node ratios above 0.17 ($p = 0.043$). In the colon cancer group, the serum BDNF concentrations significantly fell postoperatively in the entire group ($p = 0.0076$) and in subgroups of patients with or without resections ($p = 0.034$, $p = 0.0179$, respectively). Significant before-after differences were found in subgroups with angioinvasions ($p = 0.050$) and in those without neuroinvasions ($p = 0.049$). Considering the TNM stages, the postoperative BDNF concentration fell in groups with ($p = 0.0218$) and without ($p = 0.034$) distant metastases and in patients with tumours below the median size ($p = 0.018$).

**Conclusion:** Our results suggested that BDNF might play an important role in gastrointestinal cancer development. BDNF levels were correlated with tumour volume, and with neuro-, angio- and lymphatic invasions. In pancreatic cancer, BDNF concentrations varied according to the surgical procedure and they fell significantly after tumour resections. Thus, BDNF may serve as a potential marker of complete resections in underdiagnosed patients. However, this hypothesis requires

further investigation. In contrast, no differences according to the procedure was made in patients with colon cancer.

## INTRODUCTION

Oncologic diseases have become a main problem for national health services across the world. According to the World Health Organization (WHO), in 2018, there were over 18 million new cases and over 9 million deaths caused by cancer. Among these, colorectal cancer (CRC) represented 10.2% of cases and 9.2% of deaths. Although pancreatic cancer (PC) was not listed in the group of cancers with the most numerous new cases, it was counted in the group of cancers that caused the most deaths. PC caused over 430,000 deaths per year, which represented 4.5% of all deaths caused by cancer (*WHO, 2018*). Consequently, much effort has been focused on new therapies and therapeutic targets in oncologic treatment. Diagnostic methods, including new tools and substances, are constantly being evaluated for efficacy in diagnoses and follow-up monitoring. Among many others, neurotrophins seem to be a promising new target molecule in this context.

Brain derived neurotrophic factor (BDNF) belongs to the family of growth factors that includes nerve growth factor and neurotrophins 3, 4 and 5. BNDF plays an important role in brain development and neurogenesis, taking part in memory and learning processes, and it modulates energy metabolism and feeding behaviour. Serum neurotrophin concentrations have been investigated in affective disorders and cognitive impairments. Low circulating BDNF is suggested to play a protective role in neurodegeneration and stress responses and a regulatory role in metabolism (*Utami, Effendy & Amin, 2019*; *Hendriati et al., 2019*; *Munkholm, Vinberg & Kessing, 2016*; *Krabbe et al., 2009*). Cellular responses to neurotrophins are elicited *via* two categories of cell membrane receptors: low affinity $p75^{NTR}$ receptor and high affinity and specificity tyrosine kinase receptor B (TrkB). The latter one belongs to a family of cell membrane receptors which are: TrkA, TrkB, TrkC for NGF, BDNF and NT3. BDNF binding to TrkB initiates auto-phosphorylation of the receptor and initiates multiple signalling cascades, including pathways such as: the mitogen-activated protein kinase (MAPK), phosphatidyl-inositide 3-kinase (PI3K), and phospholipase C-gamma (PLC-γ). The intracellular signalling controlled by Trks and $p75^{NTR}$ is usually essential to the control of cell survival, proliferation, and differentiation. Phosphorylated TrkB activates the growth factor receptor-bound protein leading to cell proliferation and survival, metastasis formation, angiogenesis, and the inhibition of apoptosis through the MAPK and PI3K pathways. BDNF increases the expression of hypoxia-induced factor (HIF-1alfa), which upregulates vascular endothelial growth factor (VEGF) and TrkB expression, stimulating neovascularization *via* recruitment of TrkB-expressing endothelial progenitor cells (*Serafim Junior et al., 2020*).

The first described role of BDNF/TrkB in cancer was in neuroblastoma. That study showed that BDNF/TrkB expression was preferentially elevated in aggressive tumours, and

that it could promote alignancy growth, invasion, and metastasis (*Roesler, 2011*). TrkB expression in neuroblastoma was found to be associated with significantly increased expression of a subset of matrix metalloproteinases (MMPs), especially MMP-1, MMP-2 and MMP-9 which take part in extracellular matrix degradation (*Han et al., 2007*; *Chopin et al., 2016*).

The exact role of neurotrophins in oncology remains unclear. Some experimental studies have shown that overexpression of hypothalamic BDNF supports natural killer and T-cell cytotoxicity, which might be an essential anticancer mechanism. This echanism may be responsible for the positive role that BDNF displays in cancers through immunoaugmentation. Contrarily, BDNF administration resulted in changes in serum epidermal growth factor (EGF) concentrations, which facilitated the migration of lung and ovarian cancer cells (*Radin & Patel, 2017*). Moreover, the synergistic inhibition of the EGF receptor (EGFR) and TrkB suppressed colon cancer cell proliferation (*De Farias et al., 2012*). These findings might be explained by cross-talk between TrkB and EGFR signalling, which is commonly up-regulated in other cancers (*Corkery et al., 2018*; *Qiu et al., 2006*). The roles of BDNF and TrkB appear to be very important in both benign and malignant diseases. To date, however, serum neurotrophin concentrations in patients with cancer have not been widely investigated.

Here we aimed to determine the relationship between BDNF concentrations and cancer. We determined serum BDNF concentrations in patients with pancreatic or colon cancer, before and after surgery. We then analysed relationships between BDNF concentrations and different oncological parameters, such as TNM staging, lymph nodes, neural and vascular invasions, histologic grade, and tumour size.

## MATERIALS AND METHODS

### Patients

We collected serum specimens from 75 patients, 50 with CRC and 25 with PC (in the head of the pancreas). We also collected serum specimens from a control group, which consisted of 20 patients that underwent scheduled surgery, due to an inguinal hernia. We excluded patients with an inflammatory disease and patients taking immunosuppressive drugs from both the study and control groups. In all cases, the platelet counts were within the normal range. We also excluded patients with confirmed depression and patients taking psychoactive drugs during the study period, because these conditions could influence the results, as shown by *Sarabi et al. (2017)*.

All patients provided written informed consent. The study program was approved by the Ethics Committee of the Medical University of Warsaw with the number KB/56/A/2014.

### METHODS

Each patient underwent the appropriate scheduled operation, which included laparotomy, colon resection, pancreatoduodenectomy, or only a biopsy for histopathological analysis (without resection). The histopathological investigation revealed adenocarcinoma in all cases. The tumour volume was calculated by multiplying three diameters that we obtained from histopathological reports. The tumour volumes varied from 7.5 to 810 cm$^3$ in the PC

group and from 1 to 1,650 cm$^3$ in the CRC group. The median volumes were 16 cm$^3$ in the PC group and 24 cm$^3$ in the CRC group.

We measured serum BDNF concentrations, blood biochemistry, and morphology before and on the 3$^{rd}$ day after the operation. BDNF concentrations were measured with an enzyme-linked immunosorbent assay and standard chemical reagents (R&D Systems).

## Statistical analysis

Statistical analyses were performed using Statistica 13.3 (StatSoft, Poland). $P$-values less than 0.05 were considered significant.

The results were not normally distributed; therefore, we used the Mann-Whitney test for comparisons between non-related data. We performed the Wilcoxon test to compare paired groups, such as BDNF levels before and after the operation.

Patients were divided into subgroups based on the surgical procedure (resection vs. no resection), lymph node and vascular involvement (vs. no involvement), tumour volume (above vs. below the median), and the lymph node ratio (LNR; above vs. below 0.17). We set the LNR cut-off value to 0.17, based on Rosenberg et al. (2008).

## RESULTS

The study groups consisted of 50 patients with CRC (27 males and 23 females), 25 patients with PC (13 males and 12 females). The median age was 67.8 years for the entire group (CRC median age of 69.2 years, PC median age 65.4 years). The control group consisted of 20 patients (males and females) with a median age of 61.2 years.

The resection rate was 86% ($n = 43$) in CRC patients and 60% ($n = 15$) in PC patients. Clinical characteristics of study participants are shown in Fig. 1. The pre- and postoperative BDNF serum concentrations in the investigated groups and the control group are shown in Table 1. The control group showed no significant differences in serum BDNF concentrations measured before and after the inguinal hernia operation. In the CRC group, the BDNF concentration did not differ from that in the control group both before ($p = 0.5924$) and after operation ($p = 0.5399$). In the PC group, the BDNF concentration did not differ from that of the control group before operation ($p = 0.6428$) but was lower after operation ($p = 0.313$). When the PC and CRC groups were stratified by sex, the BDNF concentrations before and after operation did not differ significantly between the sexes, but men in the CRC group showed a significant reduction in BDNF after the operation (Table 1).

## Pancreatic cancer

Table 1 and Figs. 2 and 3 show BDNF concentrations before and after the operation in subgroups of patients with PC.

We found that postoperative serum BDNF concentrations (18.490 ng/ml, SD = 7.985) were significantly lower than preoperative concentrations (25.384 ng/ml, SD = 15.446, $p = 0.0455$). Among patients that underwent curative resections, the postoperative serum BDNF concentrations (16.671 ng/ml, SD = 7.612) were significantly lower than the preoperative concentrations (23.769 ng/ml, SD = 12.794, $p = 0.0098$). However, among

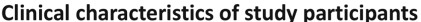

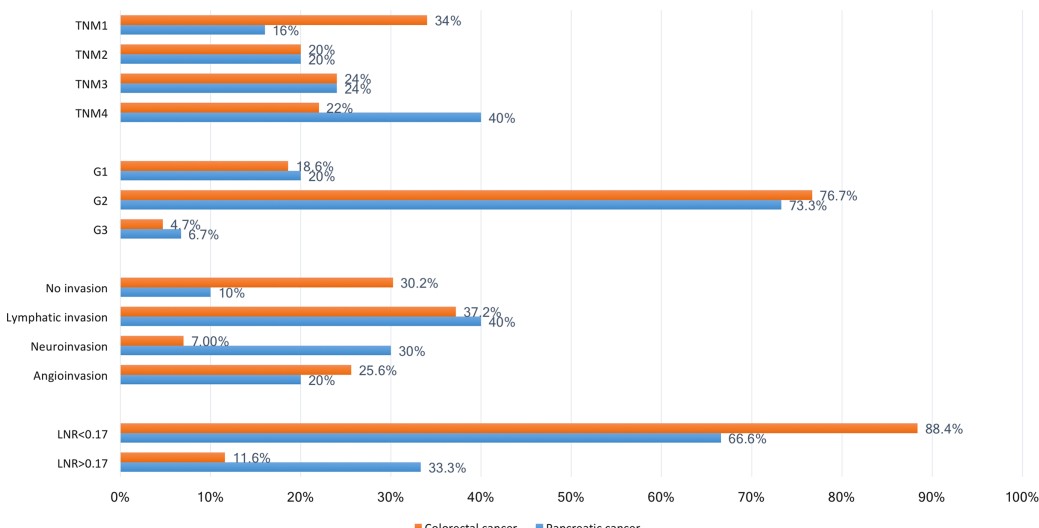

**Figure 1 Clinical characteristics of study participants.** Values are the number of patients (%) and the total number of patients analysed, unless otherwise indicated. G, histology grading; TNM, extent of invasion and/or size of tumour, regional node metastasis/involvement, metastasis; LNR, lymph node ratio, meaning the ratio of tumour-infiltrated to resected lymph nodes.

**Table 1 BDNF concentrations before and after the operation in patients with colorectal cancer, patients with pancreatic cancer, and the control group.**

| Colorectal cancer | BDNF concentration before operation (ng/ml) | BDNF concentration on 3rd day after operation (ng/ml) | p-Value* |
|---|---|---|---|
| CRC group ($n = 50$) | 26.703 ± 7.969 | 23.339 ± 9.016 | $p = 0.0076$ |
| Men ($n = 23$) | 26.549 ± 7.474 | 22.718 ± 8.994 | $p = 0.0109$ |
| Women ($n = 22$) | 24.068 ± 9.189 | 26.885 ± 8.682 | $p = 0.2123$ |
| PC group ($n = 25$) | 25.384 ± 15,446 | 18.490 ± 7.985 | $p = 0.0110$ |
| Men ($n = 13$) | 23.868 ± 11.645 | 19.081 ± 91.29 | $p = 0.0747$ |
| Women ($n = 12$) | 27.026 ± 19.155 | 17.850 ± 6.879 | $p = 0.0597$ |
| Control group ($n = 20$) | 26.413 ± 13.601 | 22.406 ± 12.060 | $p = 0.0620$ |
| Men ($n = 15$) | 22.692 ± 10.087 | 21.441 ± 9.317 | $p = 0.8927$ |
| Women ($n = 5$) | 27.754 ± 14.639 | 22.727 ± 13.120 | $p = 0.0409$ |

**Notes:**
* $p$-values less than 0.05 were considered significant.
Women and men did not show a statistically significant difference, both in the colorectal cancer and in the pancreatic cancer group.
Values are the mean ± standard deviation unless otherwise indicated.
BDNF, brain-derived neurotrophic factor; CRC, colorectal cancer; PC, pancreatic cancer.

patients that did not undergo resections, the serum BNDF concentrations were not significantly different before and after the operation (27.806 ng/ml, SD = 19.253 vs. 21.218 ng/ml, SD = 8.130, $p = 0.7518$). Similar postoperative reductions in BDNF were found in patients with 4th TNM stage tumours (without resections) and in patients with TNM-stage I, II and III tumours (with resections). Significant postoperative reductions were also noted in patients after resection of tumours below the median size (preoperative:

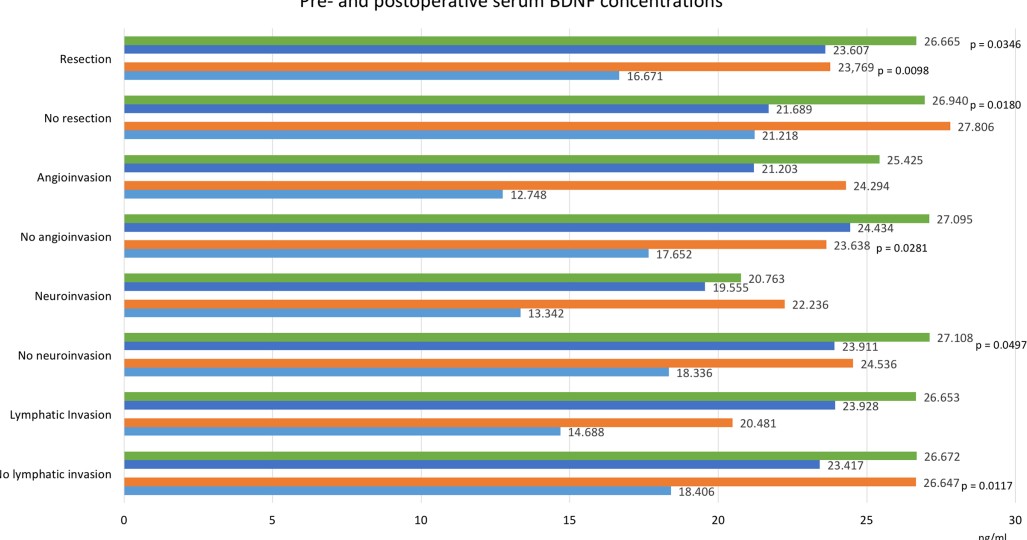

**Figure 2 Pre- and postoperative serum BDNF concentrations in patients with colorectal cancer and pancreatic cancer.** Values in ng/ml. Only significant *p*-values are indicated. CRC, colorectal cancer; PC, pancreatic cancer; BDNF, brain-derived neurotrophic factor.

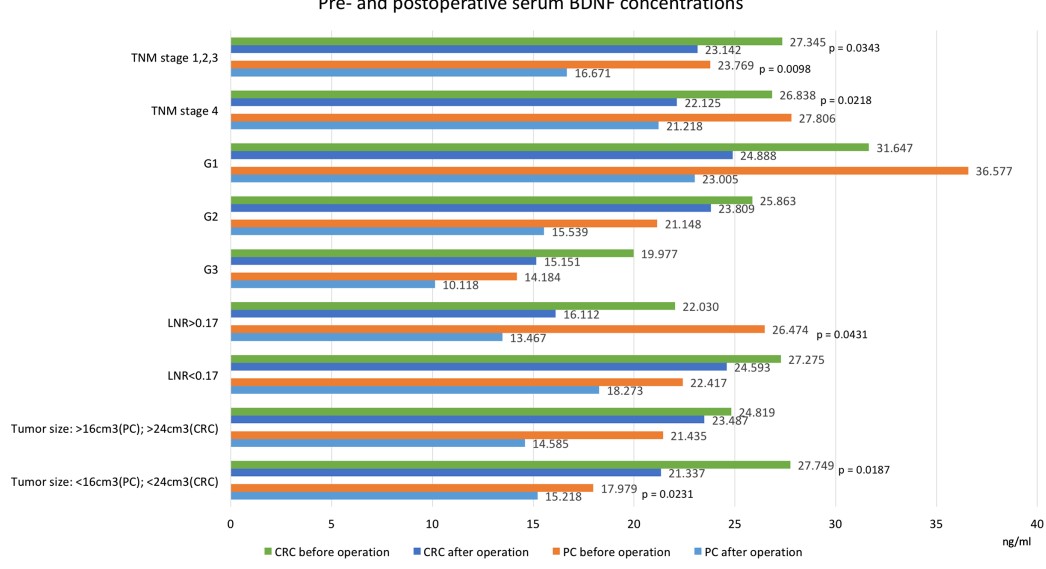

**Figure 3 Pre- and postoperative serum BDNF concentrations in patients with colorectal cancer and pancreatic cancer.** Values in ng/ml. Only significant *p*-values are indicated. CRC, colorectal cancer; PC, pancreatic cancer; BDNF, brain-derived neurotrophic factor; TNM, extent of invasion and/or size of tumour, regional node metastasis/involvement, metastasis; G, histology grading; LNR, lymph node ratio, meaning the ratio of tumour-infiltrated to resected lymph nodes.

17.979 ng/ml, SD = 7.635, postoperative: 15.218 ng/ml, SD = 4.963, *p* = 0.0231) and after resections in patients with lymph node ratios >0.17 (preoperative: 26.474 ng/ml, SD = 7.628 vs. postoperative: 13.467 ng/ml, SD = 13.331, *p* = 0.0431).

## Colorectal cancer

Table 1 and Figs. 2 and 3 show BDNF concentrations before and after the operation in subgroups of patients with CRC. When the entire CRC group was considered, we found that postoperative serum BDNF concentrations (23.339 ng/ml, SD = 9.016) were significantly lower than preoperative concentrations (26.703 ng/ml, SD = 7.969, $p = 0.0076$). The change in BDNF concentration did not depend on the type of surgical procedure; the postoperative BDNF levels were significantly lower than preoperative levels in groups with and without resections (postoperative vs. preoperative BDNF: 23.607 ng/ml, SD = 9.372 vs. 26.665 ng/ml, SD = 8.290, $p = 0.0346$, and 21.689 ng/ml, SD = 6.739 vs. 26.940 ng/ml, SD = 6.121, $p = 0.0180$, respectively). Additionally, serum BDNF concentrations declined significantly after the operations in patients with (TNM stage IV) and without (TNM stages I, II, and III) distant metastases; in patients with angioinvasions (preoperative: 25.425 ng/ml, SD = 9.489 vs. postoperative: 21.203 ng/ml, SD = 8.199, $p = 0.0505$), in those without neuroinvasions (preoperative: 27.108 ng/ml, SD = 8.201 vs. postoperative: 23.911 ng/ml, SD = 9.032, $p = 0.049713$), and in those with tumours below the median size (preoperative: 27.749 ng/ml, SD = 9.287 vs. postoperative: 21.337 ng/ml, SD = 7.843, $p = 0.0187$).

## DISCUSSION

Our results suggest that the role of neurotrophins in cancer development is essential. We confirmed that preoperative serum BDNF concentrations were significantly different from postoperative concentrations. Interestingly, in the PC group, BDNF levels only dropped postoperatively when a resection was performed. Thus, BDNF might serve as an indicator of complete curative resections for PC; however, further research is needed to test this hypothesis. To the best of our knowledge, no previous study has described this particular attribute of BDNF in gastrointestinal oncology, although its importance in oncology has been described (De Moraes et al., 2018; Tajbakhsh et al., 2017; Ozono et al., 2017).

Akil et al. (2011) showed that BDNF induces cell proliferation and has an anti-apoptotic effect mediated through TrkB, which might promote uncontrolled tumour growth. They revealed that neurotrophin and the receptor transcripts are overexpressed in tumours compared with normal tissues. Fan et al. (2014) confirmed the importance of TrkB in preventing CRC cells from undergoing anoikis, a type of apoptosis that is suggested to act as a barrier to metastasis. Silencing TrkB and BDNF inhibits the metastasis of CRC cells in vivo, and the downregulation of TrkB alone increases the sensitivity of CRC cells to anoikis in vitro. Consequently, the BDNF/TrkB pathway has been described as a potential target of anticancer therapies in patients with CRC. Radin & Patel (2017) showed that cross-talk occurred between TrkB and the EGFR signalling pathway. Thus, it has been suggested that BDNF/TrkB signalling might protect CRC cells from the antitumour effects of an EGFR blockade in anticancer therapy using EGFR-targeted monoclonal antibodies (cetuximab) (Akil et al., 2016; De Farias et al., 2012).

Other authors also report the BDNF/TrkB pathway as a very promising target for cancer therapies. In a study concerning pulmonary large-cell neuroendocrine carcinoma in a
murine xenograft model, the authors found that inhibition of TrkB signalling with the tyrosine kinase inhibitor k252a resulted in tumor regression and relapse prevention (*Odate et al., 2013*). *Thomaz et al. (2016)* investigated the effects of inhibiting TrkB signalling by use of the selective inhibitor ANA-12 in medulloblastoma. ANA -12 treatment reduced cancer cell viability, induced cell cycle arrest and, with human recombinant BDNF, reduced colony formation. Thus, TrkB inhibition might be have promise in medulloblastoma therapy.

BDNF may modulate cancer cells' sensitivity to standard drugs. In neuroblastoma, PI3K inhibition abrogates BDNF's ability to protect cancer cells from therap. The process is mediated by Bim, protein involved in intrinsic apoptosis. Decreases in Bim protected neuroblastoma cells from paclitaxel-indiced cell death (*Li et al., 2007*). BDNF might be important in maintaining resistance to apoptosis by up-regulating proteins that delay the activation of transmembrane death receptors. In breast cancer cells, production of apoptotic molecule 2 (FAIM2) through downstream of TrkB counteracts apoptosis induced by cisplatin (*Radin et al., 2016*).

A few studies have investigated serum BDNF concentrations in cancer. *Li et al. (2019)* reported that BDNF levels are lower in patients with lung cancer than in healthy controls. Those authors found higher BDNF concentrations in patients with small cell lung cancer compared with patients with non-small cell lung cancer. That finding was consistent with findings by *Kimura et al. (2018)* who noted that BDNF and TrkB expression levels are elevated in small cell lung cancer. Elevated BDNF levels are associated with cancer development and a poor prognosis. In acute pediatric leukemia, serum BDNF levels were found to be lower than in healthy controls and patients with clinical remission (*Portich et al., 2016*). The authors reported that normal BDNF levels were associated with better outcome, whereas low serum BDNF levels might serve as a potential marker of active disease. In another cancer study, in patients with CRC, serum BDNF concentrations were also found to be reduced (*Brierley et al., 2013*). In a retrospective study it was shown that BDNF concentrations were significantly lower in two cohorts of patients with CRC compared with healthy controls; however, no significant differences in BDNF concentrations between patients in different Dukes stages were found. Contrary to the above results, higher serum BDNF concentrations in CRC patients compared with healthy controls were reported by *Wang et al. (2021)*. The authors confirmed that BDNF serum levels are correlated with tumour mass and higher in metastatic patients. In the current study, we found no significant differences in preoperative serum BDNF concentrations between the cancer groups and the control group. In contrast, we found that the postoperative reduction in BDNF concentration was significantly greater in the PC group than in the control group ($p = 0.031$). We cannot explain this difference from other authors results. Significant factors which might influence BDNF and TrkB levels are cancer biology and disease advancement. In an arising tumour, due to intensive vascularisation, the serum BDNF level may be higher than it is in mature tumours where the tissue neurotrophin concentration is high. Our study confirms the significantly higher BDNF concentration in patients with TNM stage IV tumours, but we didn't find any significant differences in neurotrophin concentration related to tumour size.

There were no significant differences in preoperative BDNF levels between consecutive TNM stages in the CRC and PC groups. In the CRC group, we noted a significantly lower postoperative concentration that was not dependent on surgical procedure (*i.e.*, there was no difference between the resection and no resection groups). These results are consistent with a study by *Zoladz et al. (2019)*. Those authors investigated serum BDNF and CRP concentrations after breast surgery. They showed significantly reduced BDNF concentrations in serum (but not in plasma) at 24 h after surgery. Thus, they concluded that a decline in BDNF level might lead to an attenuation of breast cancer cell growth and metastasis (*Zoladz et al., 2019*). In the present study, in the PC group, we found significantly lower BDNF concentrations after a resection, but not after a laparotomy. No previous studies have published a similar observation. Thus, in patients with PC, the serum BDNF concentration appeared to be associated with tumour removal, because it did not significantly change when cancer tissue was not removed. This finding suggested that there might be some kind of relationship between the BDNF concentration and the presence of cancer tissues.

*Tanaka et al. (2014)* evaluated BDNF and TrkB mRNA levels in CRC, and compared their levels between primary and metastatic gastric tumours. Those authors showed that BDNF mRNA was associated with liver and peritoneal metastases. Patients that displayed overexpression of either BDNF or TrkB and patients that co-expressed both had significantly worse prognoses. Other authors have confirmed high mRNA levels of all NTs and Trks (except TrkC) in pancreatic cancer cell lines (*Ketterer et al., 2003*). Results showed significantly higher mRNA levels in cancer tissues compared with normal pancreas for BDNF but not for TrkB receptor. All NTs and their receptors were expressed at higher levels in the nerve bundles than in cancer cells, in accord with previous studies (*Sakamoto et al., 2001*). Thus, NTs might enhance cancer cells' growth and promote their proclivity to infiltrate the nerve tracts in pancreatic cancer. In an experimental study, *Fan et al. (2014)* confirmed that TrkB gene silencing and TrkB protein downregulation resulted in a slight induction of apoptosis. In a mouse model, it was shown that TrkB overexpression attenuated apoptotic trends in cancer cells.

Additionally, *Okugawa et al. (2013)* observed elevated BDNF expression in primary tumours compared with the adjacent normal mucosa in patients with gastric cancer. On the other hand, *Esfandi et al. (2019)* noted a tendency toward BDNF downregulation in gastric cancer tumour tissues. Interestingly, they found that BDNF levels were higher in tumours with lymphatic/vascular invasion, compared with tumours without lymphatic/vascular invasion, consistent with results presented previously (*Jiffar et al., 2017*; *Maehara et al., 2000*). In the present study, BDNF concentrations did not appear to be associated with lymph node involvement. We found no substantial differences in the pre- or postoperative serum BDNF concentrations between groups with and without lymphatic invasion, in either type of cancer. The only significant reduction in BDNF levels after the operation was found in the PC group with no lymphatic invasion. We evaluated the lymph node ratio (LNR) as a marker of completely resected metastatic lymph nodes, based on *Rosenberg et al. (2008)*. Those authors showed that the LNR is a strong independent prognostic factor for patients with CRC, with a cut off value of 0.17. They

showed that patients with LNRs above 0.17 have a poor prognosis. We could not confirm that finding in our analyses. In our patients with CRC, the serum BNDF concentrations were not different between groups with LNR values below and above 0.17, either before or after the operation. Interestingly, we found a significant postoperative reduction in BDNF in the PC subgroup with LNR > 0.17 ($p = 0.043$). Thus, postoperative reductions in BDNF concentrations most likely depended on lymph node status, as shown previously (*Esfandi et al., 2019*; *Jiffar et al., 2017*; *Maehara et al., 2000*).

We also divided patients into groups according to tumour size, either below or above the median tumour volume. In both cancer types, the postoperative serum BDNF levels significantly declined only in patients with smaller tumours, of below the median volume. We found no differences in preoperative BDNF concentrations based on tumour size. In some previous studies, the authors have speculated that, in CRC, larger tumours might be less aggressive than smaller tumours, and thus larger tumours might indicate a lower incidence of lymph node metastases, lymphatic invasion and venous invasion. In contrast, newly arising tumours, due to their high vascularization, might cause relevant changes in serum BDNF levels after a resection (*Fukata et al., 2020*; *Guzel et al., 2018*; *Baeten et al., 2009*). That hypothesis suggested that the BDNF concentration might not depend on tumour tissue mass; rather, it might depend on the aggressiveness of a growing tumour. Thus, high BDNF concentrations might be associated with a poor prognosis, as mentioned above. However, to our knowledge, no study has investigated the relationship between BDNF levels in cancer tissue and tumour mass.

In patients with CRC, we found a significant postoperative decline in serum BDNF concentrations when angioinvasion was present or neuroinvasion was absent. Conversely, in the PC group, a significant postoperative decline in BDNF was only observed when angioinvasion was absent. Although vascular and neural invasions were not included in the TNM staging, they both can both influence the clinical course and prognosis. In the present study, we did not observe significant differences between patients in different TNM stages, in either the CRC or PC group. However, the BDNF concentrations declined significantly postoperatively, regardless of whether patients displayed metastases. Due to the small numbers of patients in the angio/neuroinvasion subgroups, we could not determine whether angio/neuroinvasion influenced the postoperative decline in BDNF levels.

We did not observe any differences between men and women in pre- and postoperative serum BDNF concentrations. *Piccini et al. (2008)* showed that peripheral BDNF varies diurnally, and that men have significantly lower concentrations than women. Similarly, *Choi, Bhang & Ahn (2011)* confirmed that plasma BDNF concentrations underwent diurnal variations in men, but not women. However, those authors did not observe a significant difference between men and women. In the present study, we observed significant declines in BDNF serum concentrations after the operation in men, but not in women, in both the CRC and control groups. We cannot explain this result, but it might have been caused by diurnal variations in BDNF concentrations. We do not suspect that cancer had any influence on this decline.

The main limitation of this study was the small number of patients included. When the population was divided into subgroups, the numbers of participants were insufficient to draw unequivocal conclusions. Similarly, inclusion of a larger control group would make the conclusions more convincing. Assigned study not assumed measurement of TrkB and BDNF in cancer tissues what might additionally limit our conclusions, especially concerning BDNF/TrkB as a potential target for novel anticancer therapies. However, we should underline that it is a preliminary study to give a basis and direction to further research.

## CONCLUSIONS

In summary, we observed a significant connection between serum BDNF concentrations and GI cancer. Changes in serum BNDF concentrations were associated with tumour resection, although this result was significant only in the PC group. In these patients, the BDNF concentration declined significantly when the primary tumour was resected; thus, it might be considered a potential indicator of complete resection. Changes in BDNF concentration also depended on the disease stage, and they were closely related to lymph node involvement. Serum BDNF levels depended on tumour aggressiveness, rather than on tumour tissue mass.

In conclusion, it should be emphasized that our results showed that serum BDNF concentrations corresponded to the GI oncologic disease course. However, an unambiguous conclusion requires further studies on larger groups of patients.

## ACKNOWLEDGEMENTS

The authors kindly acknowledge the Department of Experimental and Clinical Pharmacology of the Medical University of Warsaw for supporting this research.

### Funding

The authors received no funding for this work.

### Competing Interests

The authors declare that they have no competing interests.

### Author Contributions

- Tomasz Guzel conceived and designed the experiments, performed the experiments, analyzed the data, prepared figures and/or tables, authored or reviewed drafts of the paper, and approved the final draft.
- Katarzyna Mech conceived and designed the experiments, performed the experiments, analyzed the data, prepared figures and/or tables, authored or reviewed drafts of the paper, and approved the final draft.
- Marzena Iwanowska conceived and designed the experiments, performed the experiments, authored or reviewed drafts of the paper, and approved the final draft.

- Marek Wroński conceived and designed the experiments, analyzed the data, authored or reviewed drafts of the paper, and approved the final draft.
- Maciej Słodkowski conceived and designed the experiments, authored or reviewed drafts of the paper, and approved the final draft.

### Human Ethics

The following information was supplied relating to ethical approvals (*i.e.*, approving body and any reference numbers):

All patients provided written informed consent. The study program was approved by the Ethics Committee of the Medical University of Warsaw (KB/56/A/2014).

### Data Availability

The raw data are available in the Supplemental Files.

### Supplemental Information

Supplemental information for this article can be found online at http://dx.doi.org/10.7717/peerj.11718#supplemental-information.

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
