# Peer review of "Brain derived neurotrophic factor declines after complete curative resection in gastrointestinal cancer"

_PeerJ, doi:10.7717/peerj.11718_

## Round 0.1 · original submission · Major Revisions

Your manuscript has now been evaluated by 3 Reviewers, all found merit in the work but raise a number of criticisms. Please address all the Reviewer's concerns in your revised manuscript and rebuttal letter.
Thank you

Reviewer 1 ·

Basic reporting

The manuscript is clearly written in professional, unambiguous language.

Experimental design

This is a well-designed study that systematically and robustly addressed a scientific hypothesis.

Validity of the findings

no comment

Additional comments

1. The article draws a conclusion that our results suggested that BDNF might play an important role in gastrointestinal cancer development. BDNF levels were correlated with tumour volume, and with neuro-, angio-, and lymphatic invasions. It may have a few deficiencies. If the author can include the serum BDNF concentration of normal people, and compare the serum BDNF concentration between the cancer patients and the normal people, this will make the article more convincing.
2. Line 78-91: more details about the role of BDNF in angiogenesis, proliferation, cell migration, and apoptosis are needed.
3. Line 90-91: There may be relevant studies about the role of serum BDNF concentration in cancer (DOI: 10.1155/2021/8867368), the author can refer to related research.
4. Line 120: Why did the author choose to detect the serum concentration on the third day after operation. Is longer time reasonable.
5. In the results section, the author mainly presents it in the form of tables. If it can be in the form of histogram or other forms, it will make the article better present the consequences.
6. All tables in the article should be a three-line table.
7. Line 287-289: Maybe the article has more limitations.
8. If the author can detect the expression of BDNF in tumor, this will make the article more convincing.

Reviewer 2 ·

Basic reporting

This manuscript is a very timely look at the serum levels of BDNf in patients with colorectal cancer. The text is clear and easy to read and comprehend. There is enough background and context provided.

Experimental design

The experimental design is suitable for the topic and type of study and the methods are easy to follow.

Validity of the findings

The findings are valid with robust underlying data provided. The conclusions are good and not over-reaching, given the small dataset.

Additional comments

An interesting and well carried out small study as a snapshot of BDNf in this patient cohort.

·

Basic reporting

The paper is adequately prepared.

Experimental design

Design, procedures and analyses are adequate.

Validity of the findings

Although it is not a large study, the number of subjects is sufficient to provide clear results.

Additional comments

Although BDNF/TrkB signaling have been increasingly investigated in cancer, the role of serum BDNF and its potential role as a biomarker in oncology is much less explored. Thus this study comparing serum BDNF levels before and after surgery in pacreatic and colon cancer will contribute to the community working on this theme and help elaborating new hypotheses for studies of BDNF as a biomarker in cancer. The number of patients could be larger but is adequate for an early study such as this one. The paper is well written and methods, result presentation data interpretation and ethical aspects are adequate.
- Lines 201 - 207, note that serum BDNF levels have been recently also measured in pediatric leukemia and low levels were found to be associated with active disease and poor prognosis, whereas normal BDNF levels were seen when the treatement was sucessful and associated wth better prognosis (Portich et al., Low brain-derived neurotrophic factor levels are associated with active disease and poor prognosis in childhood acute leukemia. Cancer Biomark. 2016 Sep 26;17(3):347-352. doi: 10.3233/CBM-160646). Can the authors speculate on how to reconcile these findings with their own findings and other published studies?
- Also note that higher tumor gene expression (transcript levels) of BDNG and NTRK2 genes have been recently associated with lower overal survival in a solid tumor type (medulloblastoma) (Thomaz et al., Neurotrophin signaling in medulloblastoma. Cancers (Basel). 2020 Sep 7;12(9):2542. doi: 10.3390/cancers12092542).
- Finally, it is important that, in the Discussion, the authors highlight the important recentl advances in the clinical use of Trk inhibition in patients with differente types of tumors harboring NTRK fusions.

---

## Round 0.2 · accepted · Accept

The Reviewers considered the paper acceptable to publication. Congratulations.

Reviewer 1 ·

Basic reporting

Authors made all requested minor revisions.

Experimental design

No comment

Validity of the findings

The results suggested that BDNF levels were correlated with tumour volume, and with neuro-, angio-, and lymphatic invasions.Thus, BDNF may serve as a potential marker of complete resections in underdiagnosed patients.

Additional comments

Authors made all requested minor revisions.